# *Prunus armeniaca* Gum-Alginate Polymeric Microspheres to Enhance the Bioavailability of Tramadol Hydrochloride: Formulation and Evaluation

**DOI:** 10.3390/pharmaceutics14050916

**Published:** 2022-04-22

**Authors:** Shazia Noureen, Sobia Noreen, Shazia Akram Ghumman, Fozia Batool, Huma Hameed, Sara Hasan, Fozia Noreen, Mervat A. Elsherif, Syed Nasir Abbas Bukhari

**Affiliations:** 1Institute of Chemistry, University of Sargodha, Sargodha 40100, Pakistan; shazianoureen11@gmail.com (S.N.); fozia.batool@uos.edu.pk (F.B.); ssarahhsn@gmail.com (S.H.); 2College of Pharmacy, University of Sargodha, Sargodha 40100, Pakistan; shazia.akram@uos.edu.pk; 3IRSET, EHSEP, INSERM, University of Rennes 1, 35000 Rennes, France; huma4748@gmail.com; 4Department of Chemistry, The University of Lahore, Sargodha Campus, Sargodha 40100, Pakistan; 5Department of Chemistry, University of Sialkot, Sialkot 51010, Pakistan; fozia.noreen@uskt.edu.pk; 6Chemistry Department, College of Science, Jouf University, P.O. Box 2014, Sakaka 72388, Saudi Arabia; maelsherif@ju.edu.sa; 7Department of Pharmaceutical Chemistry, College of Pharmacy, Jouf University, Sakaka 72388, Saudi Arabia

**Keywords:** ionotropic gelation, microspheres, polymeric blend, in vitro drug release, sodium alginate

## Abstract

Combinations of polymers can improve the functional properties of microspheres to achieve desired therapeutic goals. Hence, the present study aimed to formulate *Prunus armeniaca* gum (PAG) and sodium alginate microsphere for sustained drug release. Blended and coated microspheres were prepared using the ionotropic gelation technique. The effect of polymer concentration variation was studied on the structural and functional properties of formulated microspheres. FTIR, XRD, and thermal analysis were performed to characterize the microspheres. All the formulations were well-formed spherical beads having an average diameter from 579.23 ± 07.09 to 657.67 ± 08.74 μm. Microspheres entrapped drugs within the range 65.86 ± 0.26–83.74 ± 0.79%. The pH-dependent swelling index of coated formulations was higher than blended. FTIR spectra confirmed the presence of characteristic peaks of entrapped Tramadol hydrochloride showing no drug-polymer interaction. In vitro drug release profile showed sustained release following the Korsmeyer-Peppas kinetic model with an R^2^ value of 0.9803–0.9966. An acute toxicology study employing the oral route in Swiss albino mice showed no signs of toxicity. It can be inferred from these results that blending PAG with sodium alginate can enhance the stability of alginate microspheres and improve its drug release profile by prolonging the release time.

## 1. Introduction

Pharmaceutical technologies are still seeking novel drug delivery systems with desired objectives and minimal side effects. Recently, the fabrication of several sustained release dosages has been embedded in natural polymers because of their potential biochemical and economic advantages. These polymer-based control release devices act as reservoirs for incorporated drugs to facilitate prolonged bioavailability within the required therapeutic range with the least side effects and reduced dose frequency [1]. Many of these polymers are extracted/isolated from plant sources, including gums and mucilage. Plant gums are mainly branched polysaccharides endowed, with numerous hydroxyl groups having high water retention capability. These are hydrocolloids, hydrophilic dietary fibers capable of water absorption and hydrogels formation in low concentrations [2,3]. These polysaccharides are the preferred choice in the pharmaceutical industry due to their biocompatibility, biodegradability, and swelling capacity [4].

Many polymeric modalities have been adopted to deliver therapeutic agents at predetermined destinations in a controlled and sustained fashion [5]. Microencapsulation technology, along with prolonged drug availability, provides drug protection from hostile conditions and incompatibilities with reduced toxicity, improved efficacy, and better patient compliance [6,7,8]. Among these carrier devices, alginate-based microspheres are one of the prevalently studied carrier devices offering both controlled and targeted delivery [9,10,11]. The hydrophilic functional groups enable alginate to fabricate a porous three-dimensional architecture with phenomenal water-absorbing properties leading to controlled, modified, delayed, and extended-release formulations. However, a critical drawback of alginate microbeads is lower encapsulation and rapid degradation in an alkaline medium, causing the burst release of incorporated drugs [12]. Combining alginate with natural polysaccharides is an interesting option to overcome these shortcomings [13]. Hydrophilic matrices that blend alginate and natural polysaccharides are among the most important sustained-release oral dosages [14]. Several reported studies have revealed that blending and coating alginate with natural polymers enhanced the stability and release time of encapsulated drugs [15,16].

Physicochemical properties of Prunus gums from the Rosaceae family have been extensively studied by various researchers [17,18,19,20,21]. Hydrocolloids separated from these plants have been used in a variety of therapeutic formulations like tablets [14,22,23,24], nanoparticles [25], and mucoadhesive beads [4]. *Prunus armeniaca* gum (PAG), a high molecular weight polysaccharide, consists of L-arabinose, D-galactose, xylose, rhamnose, and mannose [19]. PAG is a promising therapeutic agent due to its expectorant, anthelmintic, and antidotal activities and is used to treat fever, lung infections, and anemia [26,27]. It is a potential emulsifier [28], antioxidant [29], stabilizer, corrosion inhibitor, and a suitable candidate for various therapeutic formulations [30]. PAG in combination with *Prunus domestica* gum extended-release time for diclofenac sodium as tablet matrix. Similarly, in another study, aceclofenac matrix tablets of PAG showed sustained release of up to 12 h. Moreover, a group of researchers observed sustained release of curcumin from PAG nanoparticles in gastrointestinal fluid up to 6 h [14,22]. PAG showed remarkable binding ability in tablet formulations [31]. Antibacterial nanoparticle of gold and silver was prepared using PAG [25]. Mucoadhesive beads of *Prunus cerasoides* gum and alginate were successfully prepared by ionotropic gellation for sustained release of diclofenac sodium [4].

Tramadol hydrochloride (TRHC) is an opioid analgesic with limited side effects than other opioid analgesics. Normally 50–100 mg dose is recommended every 4–6 h, not to exceed 400 mg per day. So a matrix system is required to modulate the release pattern of highly water-soluble TRHC to improve patient adherence and reduce the administration frequency [32]. Polymer-based microspheres are potential candidates to prolong the residence time and sustained release at the targeted site [33]. Among the various methods employed for microsphere preparation, ionic gelation is beneficial for its safety, ease, and comparatively high product yield. Variation of polymers concentration and cross-linking may be used to produce a matrix system with various drug release profiles with particular goals [6,34].

Hence the present work is intended to formulate the polymeric microsphere of TRHC using sodium alginate and PAG hydrocolloids by the ionotropic gelation method to perk up the residence time of dosage with less frequency and enhanced bioavailability. The effect of PAG and alginate proportion and preparation method (blending, coating) was examined on the stability and release behavior of formulated matrix system using Calcium chloride as a cross-linker. The effectiveness of the content variation and method applied was evaluated by various parameters like drug encapsulation efficiency, pH-dependent swelling behavior, cumulative drug release up to 12 h, and kinetic release. Structural aspects of TRHC-loaded alginate microspheres have been characterized using Fourier transform infrared (FTIR) spectra, Scanning electron microscopic (SEM) micrographs, XRD patterns, and thermograms. Acute oral toxicity of PAG-alginate was examined by histological and biochemical analysis in a mice model. The analgesic activity of microspheres was investigated using writhing response.

## 2. Materials and Methods

### 2.1. Materials

Gum was collected directly from the trunk of the *Prunus armeniaca* tree in Malakwal, Punjab, Pakistan, during July. Tramadol hydrochloride was received as a gift from Himont *Pharmaceuticals* (Pvt) Limited, Lahore, Pakistan. Sodium alginate: Mw 216 and ethanol were purchased from Sigma-Aldrich, Calcium chloride (CaCl_2_), and Potassium dihydrogen phosphate were taken from Merck. All other required reagents and chemicals were commercially available and of analytical grade. 

### 2.2. Extraction of PAG

The extraction process was adopted as reported by Fathi with minor modifications [19]. Crude gum was cleaned, crushed, and dissolved in distilled water at room temperature using a magnetic stirrer for 120 min and 160 rpm speed. Soaked gum was kept overnight at 4 °C to hydrate completely. Insoluble particles were separated through a muslin cloth. An aqueous solution obtained was centrifuged at 3000× *g* for 20 min to remove suspended particles and extracted using an ethanol solvent. The extract was dried, ground to powder, passed through an 80 mesh sieve, and saved in an airtight jar for further use.

### 2.3. Chemical Composition of PAG 

Moisture, fat, ash, and protein content were determined by AOAC standard methods [35]. Total carbohydrate content was quantified by the phenol sulfuric acid method [36]. Uronic acid quantity was determined by the carbazole method using D-galactose as the standard [37].

### 2.4. Structural Analysis of PAG

Functional groups in purified PAG powder were determined by FTIR analysis scanning between 4000–650 cm^−1^ using the KBr pellet method [38].

The crystallinity of the PAG powder was determined by XRD analysis. PXRD patterns were testified on an (X’ Pert Pro via PAN Analytical) using a monochromator CuKα radiation (1.314 A°) at 40 kV. Scanning was done over a 5° to 80° range at a scanning rate of 1°/min and a scan step of 0.05° [39]. The crystallinity index is calculated using the equation:$$\mathrm{Crystallinity}\;\mathrm{index}\;\left(\%\right)=\frac{\mathrm{Crystalline}\;\mathrm{area}}{\mathrm{Crystalline}\;\mathrm{area}\;+\;\mathrm{Amorphous}\;\mathrm{area}\;}\times 100$$

Scanning electron microscopy was used to analyze the particle shape and surface morphology of purified PAG powder. The sample was mounted on an aluminum stub using double-sided adhesive tape. The stub was coated with a thin layer of gold by sputtering. Images were captured using JEOL. JSM-5910, Tokyo, Japan, under various magnifications [40].

### 2.5. Thermal Analysis 

Thermal analysis of PAG powder was performed by an SDT Q 600 thermal analyzer. The weighed quantity of PAG was sealed in an airtight aluminum pan and heated at a rate of 5 °C/min increasing from 25 °C to 600 °C with a nitrogen gas flow rate of 20 mL/min to keep the surrounding inert [13,41].

### 2.6. Experimental Design for Preparation of Microspheres 

A three-level two-factor central composite design was used for the statistical optimization of PAG-alginate microspheres by Design Expert (version 13, Minneapolis, MN, USA). The published literature and preliminary trials were utilized to select the concentrations of PAG (X_1_) and alginate (X_2_) as independent factors, whereas the dependent variables selected were mean particle size (Y_1_), encapsulation efficiency % (EE %; Y_2_), and yield % (Y_3_). In this experiment, response surface methodology was used to optimize particle size, % EE, and % yield. A total of 13 runs was carried out to assess the influence of independent parameters on the selected dependent factors, shown in Table 1. The batches were synthesized thrice and fitted into various statistical models such as linear, 2FI, quadratic, and cubic. The fit summary comprised of adjusted and predicted R^2^ (correlation coefficient), *p*-value, and residual sum of squares was generated by the software. The model whose predicted and adjusted R^2^ are in close agreement, i.e., ≤0.2, smallest PRESS values and *p*-value ≤ 0.2 was selected accompanied by high R^2^ values. A generalized quadratic equation was used to evaluate the relationship between factors and responses [42].
Y = b_o_ + b_1 × 1_ + b_2_X_2_ + b_11_X^1^_2_ + b_22_X^2^_2_ + b_12_X_1_X_2_
where Y = response; X_1_ = PAG concentration; X_2_ = Alginate concentration; b_o_ = intercept; b_1_-b_5_= co-efficients; X^1^_2_/X^2^_2_ = second order effects of PAG and ALG concentrations respectively; X_1_X_2_ = interactive effects of PAG and ALG concentrations.

The optimized formulations were selected based on the desirability parameter and validated for the calculated statistical model by ANOVA (analysis of variance) before subjecting all to further studies [43,44].

### 2.7. Preparation of Microspheres

Suggested formulations of PAG-alginate microspheres of TRHC were fabricated by inotropic gelation using CaCl_2_ as a cross-linking agent. A summary of the preparation of microspheres is shown in Figure 1. Sodium alginate and PAG were dispersed in distilled water while stirring on a magnetic stirrer at 300 rpm. After homogenization, the pre-decided calculated quantity of PAG was slowly added to the dispersion with constant stirring. Finally, TRHC dissolved in a little quantity of distilled water was added to the polymeric blend and left for thorough mixing on a stirrer. A homogeneous blend was put drop-wise through a 24-G needle into a 10 % CaCl_2_ solution (100 mL). After incubation for a particular time, microspheres formed were separated by decantation, washed using distilled water to get rid of surface-attached drug, and dried first at room temperature and later in a desiccator to constant weight. Alginate and TRHC microspheres were prepared similarly as control.

Drug-loaded ionotropically-gelled alginate microspheres were formulated and coated by dipping in PAG solution while mild stirring for half an hour on a magnetic stirrer. The polymeric coatings were prepared with different concentrations of PAG in distilled water; 10 mL of Propylene glycol was used to improve consistency. After half an hour, coated microspheres were separated from the coating mixture, rinsed with water, and dried as blended. 

The weight of dried fabricated formulations and initially added polymers and drugs was noted to calculate percentage yield by following the formula [15].
$$\mathrm{Formulation}\;\mathrm{yield}\;\left(\%\right)=\frac{\mathrm{Weight}\;\mathrm{of}\;\mathrm{microspheres}\;\mathrm{obtained}}{\mathrm{Weight}\;\mathrm{of}\;\mathrm{polymers}+\;\mathrm{Weight}\;\mathrm{of}\;\mathrm{drug}\;}\times 100$$

### 2.8. Particle Size 

The particle size of fully dried, randomly selected 50 microspheres from each TRHC-PAG-alginate formulation was measured by an optical microscope (Olympus, Tokyo, Japan) earlier calibrated by a stage micrometer, and the average size was evaluated [45].

### 2.9. Drug Encapsulation Efficiency 

Precisely weighed 100 mg of crushed PAG-alginate microspheres were immersed into 250 mL of 7.4 phosphate buffer at 37 ± 1 °C for 24 h with occasional stirring. After 24 h, the mixture was stirred using a magnetic stirrer, and disintegrated polymeric debris was removed by filtration through Whatman filter paper (No. 40). Drug content was found using a UV–VIS spectrophotometer (Shimadzu Pharmaspec-1700, Kyoto, Japan) at 271 nm against a suitable blank. Encapsulation efficiency (%) of microspheres was calculated by following the formula [46].
$$\;\mathrm{Encapsulation}\;\mathrm{efficiency}\left(\%\right)=\frac{\mathrm{Encapsulated}\;\mathrm{drug}\;\mathrm{quantity}\;\mathrm{in}\;\mathrm{microspheres}}{\mathrm{Theoretical}\;\mathrm{drug}\;\mathrm{quantity}\;\mathrm{in}\;\mathrm{microspheres}}\times 100$$

### 2.10. Swelling and Degradation Studies 

The swelling index and in vitro degradation behavior of all formulations were found gravimetrically in different aqueous media (pH 7.4 and 1.2). 100 mg of microspheres were weighed and placed in vessels of dissolution test apparatus having 500 mL medium at 37 ± 1 °C and 50 rpm. Microspheres were removed from the medium after regular time intervals, and for swelling index wet beads were reweighted after a gentle wiping with tissue paper to remove excess water. Regarding degradation, dry weight was determined after drying in the oven. The swelling index (%) and degradation in terms of weight change were calculated using the following formulas [47].
$$\mathrm{Swelling}\;\left(\%\right)=\frac{W_{\mathrm{Swollen}}-W_{\mathrm{Initial}\;}}{W_{\mathrm{Initial}}}\times 100$$
$$\mathrm{Weight}\;\mathrm{change}=\frac{W_{\mathrm{Dry}}-W_{\mathrm{Initial}\;}}{W_{\mathrm{Initial}}}$$

### 2.11. In Vitro Drug Release Studies

TRHC release assays were executed in USP-II rotating basket-type dissolution test apparatus (Pharma max. TEST) at a rotation of 50 rpm and 37 ± 1 °C. TRHC-PAG-alginate formulations were added in 900 mL of medium (pH 7.4 and 1.2). The release was studied for two hours at pH 1.2 and for the rest of the time at pH 7.4 up to 12 h. After predetermined time intervals, 5 mL aliquots were withdrawn with immediate reinstatement of 5 mL of fresh medium to retain the similar sink conditions. Drug concentration released was observed by a UV-VIS spectrophotometer (Shimadzu Pharmaspec-1700, Kyoto, Japan) at 271 nm [48].

### 2.12. Drug Release Kinetic Analysis

Kinetics and correlation of in vitro release mechanism of TRHC from microspheres matrix need to fit into an appropriate mathematical model. Dissolution data was kinetically evaluated using different important mathematical models. Regression analysis was applied, and the accuracy and prediction capability of these models were evaluated by calculating the squared correlation coefficient (R^2^) [13,49].

### 2.13. Structural Analysis of Microspheres

Drug-polymer compatibility was studied by Fourier transform infrared spectroscope. Samples were powdered and analyzed as KBr pellets by placing them in the sample holder at 4 cm^−1^ resolution by scanning in a region of 4000–650 cm^−1^ with a speed of 1 cm s^−1^ [50].

Powder X-ray diffraction (PXRD) was executed to observe any change in the physical state of the drug during matrix formation. PXRD patterns were testified on an (X’ Pert Pro via PAN Analytical) using a monochromator CuKα radiation Cu Kα radiation (1.314 A°) at 40 kV. Scanning was done over a 5° to 70° range at a scanning rate of 1°/min and a scan step of 0.02 ° [39].

The surface morphology of TRHC-loaded alginate, PAG-alginate blended, and coated microspheres were determined by scanning electron microscope using the gold sputtering technique. Samples were coated on brass stubs with a thin film of gold under a vacuum. Morphology was examined and photographed [40].

### 2.14. Thermal Analysis

Thermal analysis of PAG-alginate microspheres was performed by SDT Q 600 thermal analyzer. Physicochemical changes in TRHC, PAG, and TRHC-loaded microspheres were evaluated by Differential scanning calorimetry (DSC). Weighed powdered microspheres were closed in an airtight aluminum pan and heated at a rate of 5 °C/min increasing from 25 °C to 600 °C with a nitrogen gas flow rate of 20 mL/min to keep the surrounding inert. Thermo-gravimetric analysis (TGA) was carried out up to 650 °C to check thermal stability [13,41].

### 2.15. Acute Toxicology Study

PAG-alginate microspheres were tested for acute toxicology by applying a Maximal Tolerance Dose (MTD) method via the oral route. Swiss albino mice (28–32 g) were purchased from the animal facility at the University of Sargodha (UOS). By following the OECD guidelines, all experiments performed were further verified by the UOS ethical committee (Ref. No. 142-2021/PREC/UOS). Animals were divided into control & treated groups with an equal number of mice (*n* = 6). All animals were fed a standard diet plus water in a clean facility under a 12 h light/dark cycle. PAG-alginate microspheres were given by oral intubation cannula method to the treated group but not to the control group. The dose of microspheres used for testing was according to excipient toxicity testing criteria. Details linked to different factors such as bad health, mortality, and any other side effect or activity were noticed for two consecutive weeks, twice daily in all animals. Biochemical and histopathological analyses were performed on all animals after the trial period. Altogether, for each parameter under this study, the treated group was compared to the control group. After finishing the trial periods, blood was collected from all mice via the orbital sinus route, and all mice were sacrificed to collect vital organs for histological studies. Separation of serum was done by centrifugation, and different clinic-pathological investigations were done, such as transaminase level (ALT and AST) and other biochemical biomarkers (cholesterol, triglycerides, creatinine, urea, and uric acid) were noted. All vital organs were weighed, and relative organ weight was calculated. All organs (heart, spleen, liver, stomach, and kidney) were preserved in formalin solution (4% buffered). After paraffin embedding, tissues were cut into 4–5 µm thickness, slides were made with cut tissue, and hematoxylin-eosin staining was done to perform histopathology analysis [51].

### 2.16. Analgesic Activity 

#### Writhing Test

A method for detecting the analgesic activity of TRHC-PAG-alginate microspheres (TPB-3) peripherally was carried out. The method was applied to overnight fasted mice by dividing them into 2 groups (*n* = 6), with free access to water. By using the intra-peritoneal route, 1% acetic acid (10 mL/kg) was given to the control group to induce writhes (contraction of abdominal muscles), and 1% acetic acid (10 mL/kg) was also given to the treated group after 2.5 h pre-administration of PAG-alginate microspheres to access the analgesic activity by counting the numbers of writhes in both groups after 5 min of a 1% acetic acid I.P injection and continued for up to 1 h [52]. A reduction in writhes counts was linked to the analgesic activity of microspheres vs. the control group. The percent inhibition of writhing was counted by the following formula;
$$\mathrm{Percent}\;\mathrm{inhibition}\;\mathrm{of}\;\mathrm{writhes}=\frac{\mathrm{Ncontrol}\;-\;\mathrm{Ntest}}{\mathrm{Ncontrol}}\times 100$$
(where N is linked to the mean number of writhes in each group)

### 2.17. Statistical Analysis

All measured data are articulated as mean ± standard deviation (SD), where *n* is the number of the data point equal to 3. Statistical and graphical analysis was performed using Origin (Origin Software, version: 2019b 9.65, OriginLab Corporation, Northampton, MA, USA). The R^2^ values of various kinetic models were calculated using DD Solver analysis. 

## 3. Results and Discussion

### 3.1. Chemical Composition of PAG

The functional properties of a gum are significantly influenced by its chemical components. Moreover, polysaccharide content reflects the purity of a gum [30]. The chemical composition of PAG is enlisted in Table 2. PAG contain 8.45% moisture, 3.86% ash, 1.26% fat, 2.53% protein, 71.32% carbohydrate, and 9.64% uronic acid. It is evident that most of the sample consists of carbohydrates, and the quantity was very close to *Prunus cerasus* gum (71.51%) and guar gum (71.1%), as reported in earlier studies [53,54]. Protein fraction can influence the emulsification and stabilization abilities of gum. The protein component of PAG (2.53%) has been found comparable to that of xanthan gum (2.125%), *Prunus dulcis* gum (2.45%), and *Soymida febrifuga* gum (2.26%) [54,55,56]. Uronic acid content is an index of acidic polysaccharides in a gum. The uronic acid content of PAG is (9.64%), higher than *Prunus cerasus* gum (7.31%) [53]. In acidic polysaccharides, carboxyl groups are dissociated at a lower pH than their dissociation constant to make them negatively charged. Interaction of these negatively charged polysaccharides with positively charged polymers and ions suggests they are a suitable candidate for encapsulating ingredients [30]. Ash content of PAG is 3.86% which is close to *Albizia procera* gum (4.1%) but higher than gum Arabic (1.2%) [54,57].

### 3.2. Structural Analysis of PAG 

The FTIR spectrum of PAG powder is depicted in Figure 2a, which shows all typical peaks associated with polysaccharides. Peaks at 3636.01 cm^−1^ and 3259.55 cm^−1^ are due to OH bond stretching [58]. The absorption band at 2924.09 cm^−1^ and 2888.68 cm^−1^ are attributed to the CH bond stretching of CH_3_ and CH_2_ of groups [59]. A small peak at 2161.85 cm^−1^ is due to C=C bond vibration [60]. Two peaks at 1735.07 cm^−1^ and 1597.16 cm^−1^ are attributed to the C=O stretching of the COOH group [19]. Two other peaks at 1457.38 cm^−1^ and 1418.25 cm^−1^ correspond to CH bending in CH_3_ and CH_2_ monosaccharide units [60]. The band at 1289.65 cm^−1^ and 1136.83 cm^−1^ are due to stretching movements of the pyranose ring [61]. Another absorption peak at 1002.65 cm^−1^ may refer to CN stretching [19]. A peak at 864.74 cm^−1^ indicates α-linkage in sugars [61]. It can be inferred from IR spectral analysis that several hydrogen bond-forming groups are present in PAG polymer, which suggests good bioadhesive nature and viscous formulation development [62].

XRD results revealed (Figure 2b) that PAG is amorphous with a partially crystalline structure. The crystallinity index was 35.71%. Almond gum and *Prunus domestica* gum have shown similar structures [63,64]. Highly amorphous polysaccharides are advantageous to use in the development of drug delivery systems due to their higher solubility [65]. 

Particle shape and surface morphology greatly influence the water absorption ability of a gum, which is further related to intrinsic viscosity and molecular mass [40]. The PAG particles have irregular size and shape (Figure 3a) and rough surfaces (Figure 3b,c). The microphotographs of the gum showed an amorphous nature supported by XRD analysis. The irregular shape and rough surface increase the possible voids between particles and, in turn, influence the gum density [20].

### 3.3. Thermal Analysis

The thermal analysis is used to understand the stability of gum and gum-based products at elevated temperatures [55]. The Thermogram of PAG (Figure 4a) shows three subsequent weight losses with increased temperature. Initially, a weight loss of 17.32% from 30 to 242.94 °C was observed due to the dehydration of the gum. Second, weight loss up to 306.81 °C can be attributed to the thermal degradation of the polymer. Third, weight loss indicates the conversion of PAG polymer into carbon residues. Similar TGA results have been reported for thermal analysis of Acacia gum, *Dalbergia sissoo* gum, and almond gum [20,63,66]. DSC curve of PAG (Figure 4b) showed an endothermic transition at 480.86 °C running through a peak of 499.96 °C with an enthalpy change of 3.244 J/g. Thermal analysis results suggest that PAG is a thermostable polymer that could be used to develop a variety of biomaterials [62].

### 3.4. Experimental Design and Response Surface Methodology

A key parameter in the drug delivery process is the efficient improvement and optimization of the experimental design while magnifying the practical efficiency. For this, CCD has proven to be essentially useful, which effectively overcomes the deficiencies in the optimization routes. In the present study, the formulation and optimization of PAG-alginate microspheres were carried out by a 2-factor 3-level CCD response surface methodology (RSM). The independent factors viz PAG and alginate concentrations were quantitatively assessed to evaluate their effect on dependent responses, i.e., mean particle size, encapsulation efficiency percentage, and yield %. 

The ANOVA (analysis of variance) predicted the significant effect of the PAG-alginate concentrations on all responses with *p* ≤ 0.01. The dependent variables, namely PS, EE, and yield, were fitted into regression analysis by comparing linear, 2FI, quadratic and cubic models. The quadratic model was chosen as the best fit model for the experiment for all responses by evaluating several statistical factors, namely R^2^, correlation co-efficient, predicted and adjusted R^2,^ and PRESS (predicted residual sum of squares), shown in Table 3 Furthermore, the justification of adequacy of each model for every response was made by F-value and *p*-values evaluated by ANOVA (Table 3) [66]. The quadratic model was found to have the lowest PRESS value, shown in Table 3. Among the statistical model, the quadratic model was found to be statistically significant with F values (15.67, 31.94, and 5.98) and *p*-values (0.01, 0.003, 0.05) for PS, EE (%), and yield (%), respectively. Moreover, the reliability of the statistical model applied was also checked by comparing adjusted and predicted R^2^ values for all responses. The adjusted R^2^ values evaluated were close to the predicted R2 (Table 3). The difference between adjusted and predicted R^2^ values was ≤0.2 for all response variables, signifying the signal/noise ratio [42]. This also advocated the adequacy and significance of regression model equations to explain the relationship between independent and dependent variables [67]. The justification of the adequacy of each model for every response was made by correlation coefficient R^2^, F-value, and *p*-values evaluated by ANOVA (Table 3) [66]. Among the statistical models, the quadratic model was found to be the most statistically significant, with F values (15.67, 31.94, and 5.98) and *p*-values (0.01, 0.003, 0.05) for PS, EE (%), and yield (%), respectively. Moreover, the adjusted R^2^ values viz; 0.9933, 0.8633, and 0.9896 were closed to predicted R^2^ with an error ≤ 0.2, advocating the adequacy and significance of regression equations to explain the relationship between independent and dependent variables [67]. The second polynomial equations obtained were used for describing the mathematical relation between independent and dependent factors (Equations (1)–(3)).
Mean particle Size = 606.8 + 29.47X_1_ + 12.65X_2_ + 15.24X_1_X_2_ + 19.52X_1_^2^ − 0.833X_2_^2^(1)
% EE = 82.79 + 3.83X_1_ + 2.26X_2_ − 3.4X_1_X_2_ − 2.85X_1_^2^ − 2.52X_2_^2^(2)
% Yield = 91.31 + 1.30X_1_ + 0.08X_2_ − 10.29X_1_X_2_ − 8.51X_1_^2^ − 3.42X_2_^2^(3)

The magnitude and sign of coefficient in a polynomial equation are also important as these describe the effect of independent factors on response. The coefficient gives us a response amount that changes upon altering the coded parameters while keeping the others the same. The sign elucidates the change in dependent responses with a positive sign indicating a synergistic/positive effect and a negative sign showing an antagonistic effect. The positive sign in Equations (1)–(3) with X_1_ and X_2_ shows that as the concentration of PAG and ALG increases, the particle size, %EE and % yield increase [68].

Additionally, the actual and predicted values plotted by linear correlation graphs were close on a straight line, shown in Figure 5. The graph showed a high fitting degree of the model, confirming the accuracy of the data, thus affirming the model’s suitability for simulating the experiment data for PAG-alginate microspheres.

The 3D response surface plots and 2D contour graphs shown in Figure 5 were constructed to elaborate on the effect of selected parameters on dependent responses. The graph evidenced that by changing the polymer concentrations, the parameters significantly changed. The increase in EE and yield by increasing PAG; ALG ratio could be owed to the increased medium’s viscosity which rapidly solidifies the microspheres reducing the drug diffusion from the matrix. The PS was substantially reduced by ALG concentration increase, whereas PAG amount increase showed the opposite effect on PS [69]. 

### 3.5. Formulation Optimization

A predictable optimization technique by desirability method was used to develop optimized formulations. After preliminary testing, the desired parameters were set to obtain the microsphere size ranging within 500–700 µm with maximal % EE and maximum % yield. The optimized microspheres were formulated with 4:2 and 3:3 ratios of ALG and PAG, respectively. Their response values were the closest to the predicted ones, with the desirability of 0.811 and 0.842, shown in Table 1 [66]. 

### 3.6. Preparation of Microspheres

Alginate, a naturally occurring anionic polysaccharide, forms gel beads through sol-gel transition in the presence of divalent cations like Ca^2+^ and Zn^2+^ and plays a promising role in drug delivery [70]. Alginate microspheres are stable in an acidic medium, but erosion in an alkaline medium due to electrostatic repulsion between carboxylate ions after replacing calcium ions with sodium makes it a poor choice for sustained drug release [16]. The blending of polymers can improve functional properties and achieve required therapeutic goals. Additionally, various polymer ratios can give more acceptable physicochemical properties with improved drug release patterns. A pharmaceutical product with desired qualities needs to find an appropriate combination of variables [50,71]. The critical variables of ionotropic gelation are a drug-polymer composition of formulations and cross-linking conditions [34]. The effect of variations of polymer components and preparation methods (blending and coating) on drug release was studied, keeping drug ratio and cross-linker concentration constant. TRHC containing polymeric suspension immediately produced microspheres in CaCl_2_ solution. Divalent calcium ions cross-linked two alginate chains by attaching ionically to their carboxyl group [72]. All formulations were well-formed, spherical in shape, and stable, with good yields from 80.46 ± 1.08–91.27 ± 0.85. Blended microspheres were slightly darker in color than sole alginate drug-loaded beads. Various combinations of TRHC, alginate, and PAG are prepared, and their yield is given in Table 4.

### 3.7. Particle Size

Interaction of alginate with divalent calcium ion in CaCl_2_ solution cross likened carboxylate group of alginate and resulted in the formation of spherical, stable PAG-alginate beads with rough surfaces [73]. The average diameter of TRHC-PAG-alginate microspheres was within the range of 579.23 ± 07.09 μm to 657.67 ± 08.74 μm (Table 4). Coated microspheres were smaller in size than blended microspheres. 

### 3.8. Drug Encapsulation Efficiency 

Drug encapsulation efficiency is predominantly affected by physicochemical and architectural properties of the polymeric system and drug-polymer interactions and also reflects the efficiency of the preparation technique to incorporate selected drugs [74]. The entrapment efficiency of TRHC-PAG-alginate microspheres was 65.86 ± 0.26–83.74 ± 0.79, as shown in Table 4. Generally, blended formulations encapsulated more drugs as compared to coated. The lower drug entrapment efficiency of coated formulations was due to the diffusion of the drug from microspheres during the coating process [16]. It was noted that the encapsulation efficiency of blended microspheres increased by increasing PAG polymer concentration. This increase might be attributed to the enhanced viscosity of the polymer matrix due to a higher concentration of PAG [75]. 

### 3.9. Swelling and Degradation Studies 

Water absorption capacity is an important attribute of the drug delivery system, which leads to matrix swelling, drug diffusion, polymer network degradation, and mucoadhesion [6]. The swelling behavior of microspheres was evaluated in a gastric acidic (pH 1.2) and phosphate buffer (pH 7.4) at 37 °C. Ionotropicaly cross-linked alginate polymer matrices were pH-responsive [75]. The swelling index was significantly lower in pH 1.2 buffers, and microspheres were stable in an acidic medium, as reported earlier (Figure 6a) [76,77]. Lower swelling and stability of PAG-alginate microspheres in an acidic environment are most likely due to the replacement of protons with calcium ions to form insoluble alginic acid pursued by solvent penetration [78]. In pH 7.4, all types of microspheres showed significantly higher swelling (Figure 6b). This was due to the relaxation of polymeric chains and deprotonation of carboxylic acid groups in the basic medium [77,79]. In 7.4 pH alginate microspheres, TA-1 showed maximum swelling up to 6 h and started to disintegrate [16]. PAG-alginate microspheres showed maximum swelling up to 8 h and started to degrade gradually. The swelling index was higher for coated formulations than for blended. Moreover, swelling was increased by increasing the concentration of coating. PAG incorporation decreased swelling index but increased time to maximum swelling [4].

As indicated by weight loss (Figure 6c), PAG-alginate microspheres showed a slow degradation than sole alginate microspheres. In pH 7.4 phosphate buffers, there was a rapid degradation observed in alginate microspheres after 2 h, and most of the beads were disintegrated up to 5 h. Both blended and coated PAG-alginate microspheres were relatively stable [47]. Blended microspheres were degraded slowly and then coated. Maximum coated microspheres lost their spherical shape within 12 h. While a reasonable fraction of blended microspheres was still in its shape after the same interval of time. There was no significant weight change in pH 1.2. These results indicate that polymer-alginate combinations are more stable in pH 7.4 than sole alginate microspheres [15].

### 3.10. In Vitro Drug Release Studies

Drug release from microspheres depends upon the morphology, size, degree of cross-linking, pH of dissolution media, and physicochemical properties of the drugs [80]. In vitro drug release patterns of all formulations were studied, and graphs were plotted for cumulative drug release percentage. It can be inferred from Figure 7 that a very limited fraction of TRHC was released up to 2 h in an acidic medium (pH 1.2), probably due to drug present at the surface of microspheres. In an acidic medium, alginate show poor solubility hence lower swelling and drug release. In phosphate buffer (pH 7.4) for the rest of the time, higher swelling of PAG-alginate microspheres facilitates more drug release [13,49]. Alginate TRHC microspheres TA-1 released 94.07 ± 3.34% drug till 8 h. It may be due to the replacement of calcium ions with sodium ions in a medium resulting in the repulsion in polymer chains and, finally, erosion [16]. Blended microspheres TPB-2 and TPB-3 were relatively stable, swelled, and released the drug up to 80.62 ± 4.80% and 75.57 ± 3.45% till 12 h. It can be observed that blending PAG polymer with alginate increased the duration of drug release [50,81]. Higher drug release from blended microspheres may be due to the gradual swelling of PAG-alginate beads in pH 7.4 solution [73]. Coated microspheres TPC-4 and TPC-5 released drugs up to 72.88 ± 4.98% and 70.37 ± 3.49% till 12 h. This means coating alginate with PAG polymer retarded the release of TRHC from microspheres. An increase in the coating concentration slowed the release of drugs [15]. Polymer coating strengthens the alginate matrix and decreases the permeability of the surface layer [16]. 

### 3.11. Drug Release Kinetic Analysis

The dissolution data of various TRHC-PAG-alginate formulations were fitted to various kinetic models like Zero Order, First order, Higuchi, and the Korsmeyer-Peppas model. Analysis of release data modeling revealed that TRHC-loaded microspheres followed the Korsmeyer-Peppas model. Results of the release curve fitting into different mathematical models are given in Table 5. These results indicated the best fitting of the Korsmeyer-Peppas model for drug release (R^2^ = 0.9803–0.9966) with a release exponent value *n* 1.434–2.041. Besides this Zero-order model was also observed (R^2^ = 0.8467–0.9451) to fit Korsmeyer-Peppas model best. Best fitting to the Korsmeyer-Peppas model suggested that drug release from drugs released from various PAG-alginate formulations followed a controlled release pattern [15,82].

### 3.12. Structural Analysis of Microspheres

In the pure TRHC spectrum (Figure 8), characteristic peaks appeared at 3300.55 cm^−1^ for OH and 2929.68 and 2860.73 cm^−1^ for CH stretching. An evident distinctive fingerprint region appeared between 1600 and 650 cm^−1^ [83]. The PAG area between 1200–800 cm^−1^ is characteristic of carbohydrate structures. A peak at 1735.07 cm^−1^ refers to the COOH group, and absorption at 1457.38 cm^−1^ and 1418.25 cm^−1^ may be due to CH_3_ and CH_2_ bond vibrations. A peak at 3259.55 cm^−1^ can be associated with OH stretching. Another peak at 2888.68 cm^−1^ may be due to CH stretching [19]. IR spectra of blended microspheres (TPB-3) showed a peak at 3300.55 cm^−1^ and coated microspheres (TPC-4) at 3311.74 cm^−1^ for OH stretching while at 2935.27 cm^−1^ (TPB-3) and 2916.64 cm^−1^ (TPC-4) for CH stretching. As the characteristic peaks of TRHC remained unchanged, so there were no drug-polymer interactions in formulations [84].

Powder X-ray crystallography of TRHC illustrated characteristic sharp peaks at diffraction angles between 10° to 30°. In the case of TRHC-loaded blended and coated PAG-alginate microspheres, no such peaks were observed at these angles (Figure 9). These results proposed that the drug remained in its amorphous form in the polymer matrix [85,86]. Conversion of crystalline form into amorphous may be due to conditions applied for microsphere preparation [87].

The micrographs of the TRHC-loaded sole alginate and PAG-alginate blended and coated microbeads were well-rounded spheres (Figure 10). Drug-loaded alginate and PAG blended microspheres have a uniformly rough, somewhat porous and wrinkled surface. These small pores and wrinkles may be due to squeezing out of water during drying. Uniform rough surface may be due to the interaction of polymer-polymer and/or polymer cross-linkers [13,88].

### 3.13. Thermal Analysis

To confirm the physical state of TRHC in PAG-alginate formulation, DSC of pure TRHC, PAG, and TPB-3 was performed (Figure 11a). DSE of the pure drug showed a sharp endothermic peak at 182.9 °C, indicating the melting of the drug [89]. No such peak was observed in the DSC of TRHC-PAG-alginate microspheres which confirms that drug-loaded microspheres changed into amorphous form during the formation process [87]. 

An overlay of TGA results of TRHC, PAG, and loaded PAG-alginate microspheres is shown in Figure 11b. TGA curve of pure TRHC shows that the drug is stable up to 200 °C [89], and maximum weight loss occurred between 200 °C to 250 °C as about 90% mass was lost. TGA curve of PAG showed that gum polymer was stable up to 243 °C, and there was a gradual mass loss up to 500 °C with a maximum loss between 250 °C to 300 °C near 30%. In the TGA curve of TRHC-PAG-alginate an initial mass loss was observed at 72 °C due to moisture loss [90]. After removing moisture, a mass loss was observed near 200 °C. Above 200 °C formulation was stable, and a very slow mass loss was observed up to 600 °C. These results indicate the thermal stability of formulated TRHC- PAG-alginate microspheres.

### 3.14. Acute Toxicology Study

#### 3.14.1. General Observations

No death, mortality, or illness-linked signs were seen in both control and treated groups over the trial period (2 weeks). Overall, weight-linked changes were insignificant in both groups (Table 6). All vital organs (liver, stomach, heart, kidney, and spleen) remained unaffected in both groups. The effects of oral intake of TRHC-PAG-alginate linked to different factors such as food, water intake, and body weight were also recorded and mentioned in Table 6. Food and water intake was a little less in the treated group than in the control group (which May be linked to the fullness of the stomach via the administration of microspheres). Overall, food and water consumption in both groups were seen as casual. LD50 value (>2000 mg/kg) in Swiss albino mice and toxicity score was found in accordance with a globally harmonized system [51]. Therefore, our TRHC-PAG-alginate formulation was in agreement with the category mentioned above, having a zero toxicity score.

#### 3.14.2. Biochemical Analysis

Biochemical analysis was done using serum to see differences between physiological and pathological conditions. Different chemical parameters linked to liver, lipid profiles, and kidney functioning were found in both groups and mentioned in Table 7. Transaminases level was found in the reference range (28–180 IU/L) in both groups. Other parameters such as lipid profiles (Cholesterol and Triglycerides level), creatinine, urea, and uric acid levels were also in the reference range. Altogether, no sign of toxicity was found.

#### 3.14.3. Histological Analysis

The absolute weight of each organ was noted in both groups, and the mean weights in both groups are mentioned in Table 8. No changes linked to histology were seen in the liver, stomach, heart, spleen, and kidney, indicating the safety of TRHC-PAG-alginate in any sort of significant pathology (Figure 12).

### 3.15. Analgesic Activity Evaluation

Analgesic activity was determined by the writhing response in both groups. A reduction in the number of writhes induced by acetic acid was seen at around 50% in the mice group treated with drug-loaded PAG-alginate microspheres compared to non-drug-loaded PAG-alginates microspheres, as shown in Table 9. Results were mentioned as mean ± SEM in each group (*n* = 6). Statistical mean differences between control and treated groups were assessed using the non-parametric Mann-Whitney U-test (Significance: ** *p* < 0.01). Results of analgesic activity are given in Table 9.

## 4. Conclusions

TRHC-loaded microspheres were prepared using sole alginate and various combinations of alginate and PAG by blending and coating using the ionotropic gelation technique to prolong the release time of the drug. Microspheres were well-formed spherical particles with a rough surface and high yield. Drug entrapment efficiency was higher in blended formulation than in alginate and coated batches. Swelling behavior and release pattern were pH-dependent. PAG-alginate microspheres were more stable in phosphate buffer than alginate microspheres and exhibited extended residence time for the loaded drug compared to alginate-based microspheres. The retention time of TRHC in the case of coated microspheres was higher, but the encapsulation efficiency of coated microspheres was lower. Kinetics evaluation suggested the Korsmeyer-Peppas model for release profile, indicating a controlled release system. XRD indicated the amorphous nature of TRHC in microspheres which was further confirmed by DSC. The acute oral toxicity assessment of the optimized formulation showed no toxicity signs or histological changes. It is evident from results that incorporation of PAG has increased the stability of alginate microspheres which in turn prolonged the availability of TRHC. So, these formulated polymeric microspheres may be a potential matrix for sustained drug delivery as a therapeutic alternative.

## Figures and Tables

**Figure 1 pharmaceutics-14-00916-f001:**
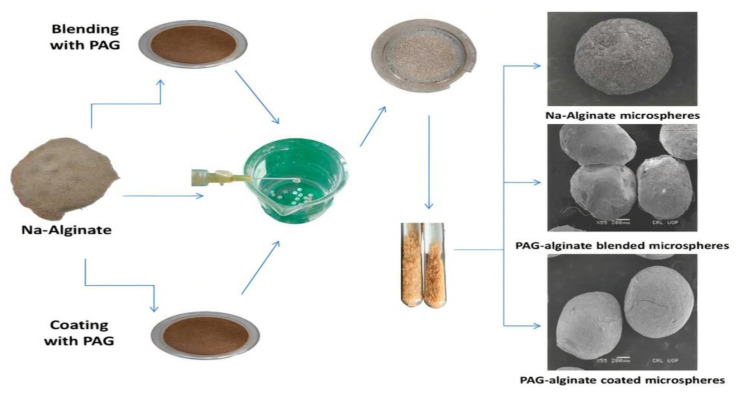
Schematic diagram for preparation of TRHC-loaded alginate and PAG-alginate blended and coated microspheres.

**Figure 2 pharmaceutics-14-00916-f002:**
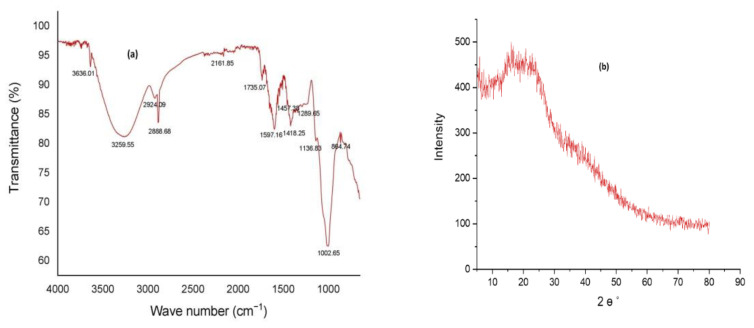
(**a**) FTIR spectra of purified PAG; (**b**) XRD of pure purified PAG.

**Figure 3 pharmaceutics-14-00916-f003:**
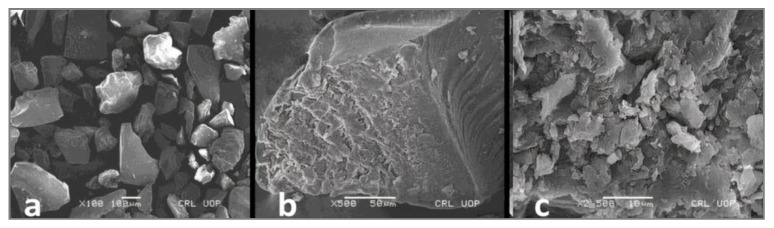
Scanning electron microphotograph of purified PAG; (**a**) shape of PAG particles; (**b**,**c**) surface morphology of PAG particles.

**Figure 4 pharmaceutics-14-00916-f004:**
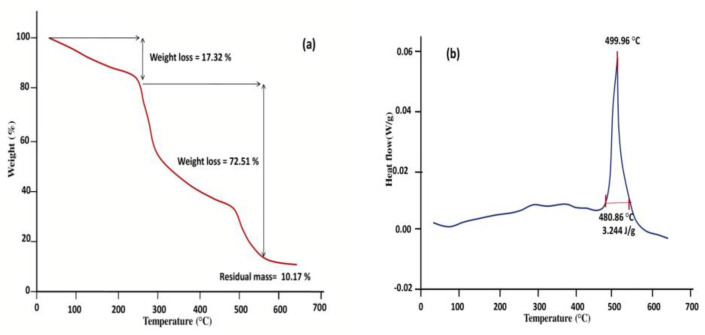
Thermal analysis (**a**) TGA, (**b**) DSC thermograms of purified PAG.

**Figure 5 pharmaceutics-14-00916-f005:**
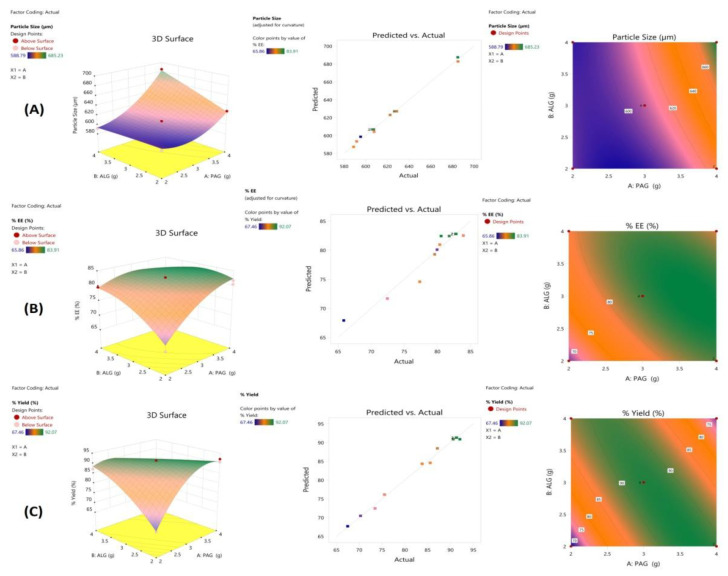
2D contour graphs, predicted vs. actual value and 3D response surface plots for; (**A**) P.S. (particle size); (**B**) EE % (Encapsulation efficiency); (**C**) Yield %.

**Figure 6 pharmaceutics-14-00916-f006:**
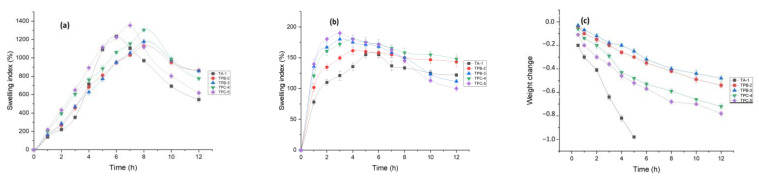
Swelling index (%) of TRHC-PAG-alginate microspheres: (**a**) pH 1.2; (**b**) pH 7.4; (**c**) weight change in pH 7.4 (Mean ± SD, *n* = 3).

**Figure 7 pharmaceutics-14-00916-f007:**
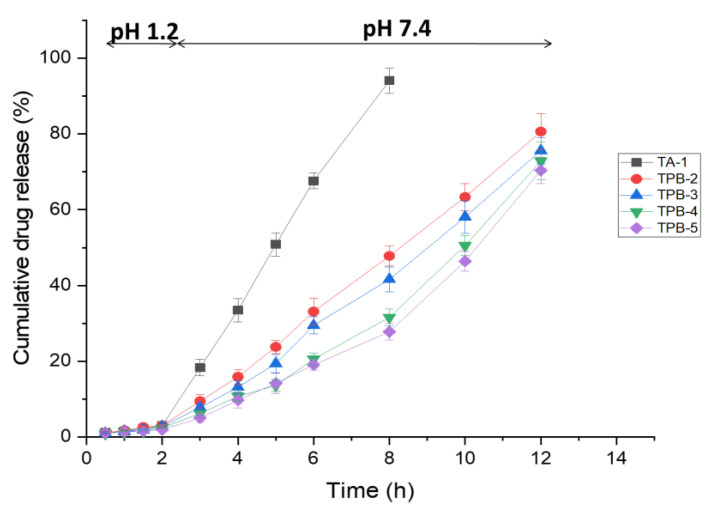
Cumulative drug release % of TRHC-PAG-alginate microspheres from 1–2 h in pH 1.2 and 3–12 h in pH 7.4 (Mean ± SD, *n* = 3).

**Figure 8 pharmaceutics-14-00916-f008:**
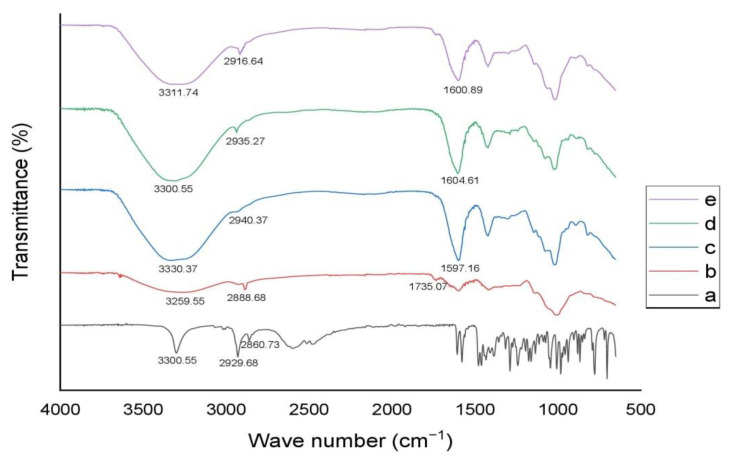
FTIR spectra of (**a**) TRHC (**b**) PAG (**c**) unloaded PAG-alginate microspheres (**d**) blended microspheres (**e**) coated microsphere.

**Figure 9 pharmaceutics-14-00916-f009:**
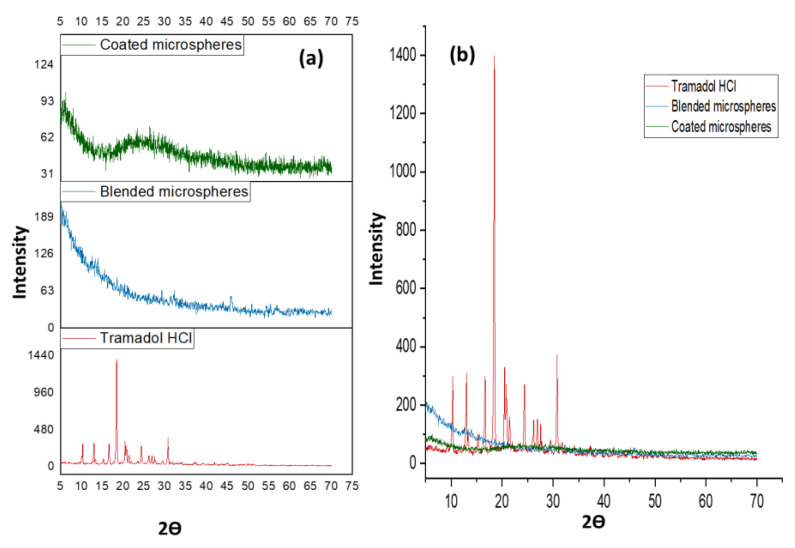
(**a**) XRD of pure TRHC, coated microspheres, blended microspheres; (**b**) Overlays of diffractograms showing a change in the intensity of peaks of TRHC in microspheres.

**Figure 10 pharmaceutics-14-00916-f010:**
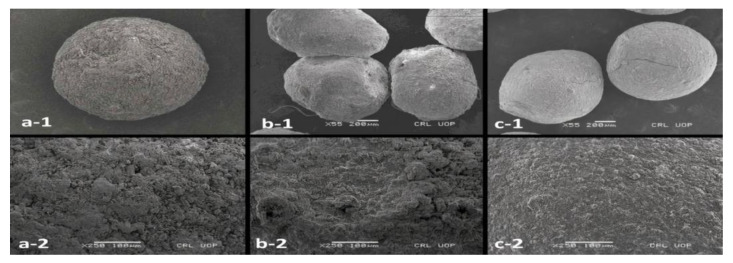
Scanning electron microphotograph showing: (**1**) shape; (**2**) surface morphology; (**a**) alginate microspheres; (**b**) blended microspheres;(**c**) coated microsphere.

**Figure 11 pharmaceutics-14-00916-f011:**
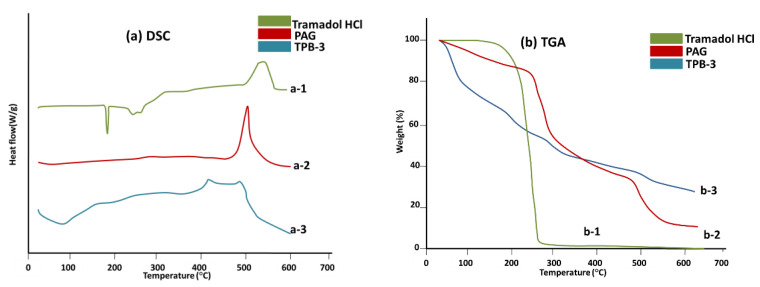
Thermal analysis: (**a**) DSC thermograms of(a-1), pure TRHC; (a-2) PAG and (a-3) TRHC loaded PAG-alginate microspheres (**b**) TGA curves of (b-1) pure TRHC; (b-2) PAG; and (b-3) TRHC loaded PAG-alginate microspheres.

**Figure 12 pharmaceutics-14-00916-f012:**
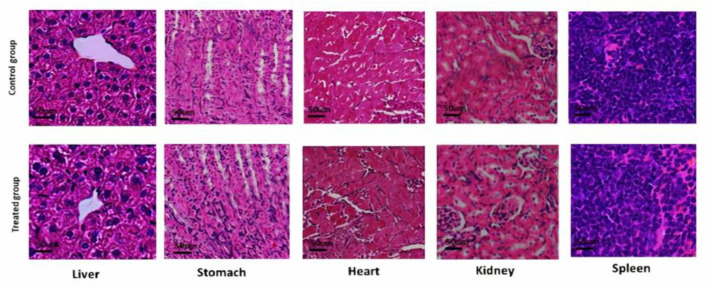
Histological analysis of control and TRHC-PAG-alginate treated group.

**Table 1 pharmaceutics-14-00916-t001:** Central composite design of two independent factors with actual and predicted values of response variables.

Formulation Code	Concentration (g)		Responses					
	PAG	ALG	PS (µm)		EE (%)		Yield (%)	
	X_1_	X_2_	Y_1_		Y_2_		Y_3_	
			Exp.	Pred. *	Exp.	Pred. **	Exp.	Pred. ***
1	4	2	626.42	627.07	81.78	82.42	90.60	90.89
2	4.4	3	685.00	687.52	83.91	82.50	75.54	76.13
3	2	4	591.54	593.42	79.59	79.27	87.13	88.45
4	3	4.4	622.45	623.01	80.36	80.94	85.56	84.58
5	3	3	607.34	606.80	82.74	82.79	91.27	91.31
6	4	2	628.62	627.07	80.56	82.42	92.07	90.89
7	4	4	685.23	682.84	79.95	80.08	70.28	70.47
8	3	3	606.25	606.80	82.83	82.79	91.34	91.31
9	1.5	3	607.67	604.16	72.45	71.67	73.45	72.44
10	3	1.5	588.79	587.24	77.33	74.56	83.78	84.34
11	2	2	595.23	598.60	65.86	67.91	67.46	67.69
12	3	3	607.34	606.80	82.74	82.79	91.27	91.31
13	3	3	606.25	606.80	82.83	82.79	91.34	91.31

PAG: *Prunus armeniaca* gum; ALG: Alginate; P.S.: Particle size; EE: Encapsulation efficiency; Exp.: Experimental; Pred.: Predicted * Predicted response of particle size was estimated by second-order polynomial Equation (1). ** Predicted response of EE was estimated by second-order polynomial Equation (2). *** Predicted yield response was estimated by second-order polynomial Equation (3).

**Table 2 pharmaceutics-14-00916-t002:** Chemical composition of purified PAG.

Component	Quantity (%)
Moisture	8.45 ± 0.29
Ash	3.86 ± 0.06
Fat	1.26 ± 0.09
Protein	2.53 ± 0.05
Total sugars	71.32 ± 0.78
Uronic acid	9.64 ± 0.21

**Table 3 pharmaceutics-14-00916-t003:** **A**. Fit Summary for regression model selection. **B**. ANOVA results for regression model equations.

A									
Factors	Source		Sequential *p*-Value	Lack of Fit *p*-Value	Adjusted R2	Predicted R2	PRESS	Remarks	
Mean Particle Size (µm)	Linear		0.002	<0.0001	0.632	0.395	7176.4	--------	
	2FI		0.104	<0.0001	0.7	0.542	5418.13	--------	
	Quadratic		<0.0001	0.0112	0.993	0.975	285.09	Suggested	
	Cubic		0.006	0.1667	0.998	0.987	146.13	Aliased	
Encapsulation efficiency (%)	Linear		0.022	0.0002	0.439	0.131	269.40	--------	
	2FI		0.067	0.0003	0.579	0.244	234.34	--------	
	Quadratic		0.003	0.003	0.862	0.684	97.96	Suggested	
	Cubic		0.126	0.0032	0.936	−0.276	396.03	Aliased	
Yield (%)	Linear		0.828	<0.0001	−0.155	−0.829	1791.02	--------	
	2FI		0.019	<0.0001	0.320	−0.054	1032.84	--------	
	Quadratic		<0.0001	0.0584	0.989	0.965	33.75	Suggested	
	Cubic		0.042	0.2106	0.995	0.963	35.56	Aliased	
**B**									
**Terms**	**Responses**								
	**P.S.**			**EE (%)**			**Yield (%)**		
**Source**	**Sum of Squares**	**F-Value**	***p*-Value**	**Sum of Squares**	**F-Value**	***p*-Value**	**Sum of Squares**	**F-Value**	***p*-Value**
Model	11,803.49	359.13	<0.0001	291.5	21.74	0.0004	973.09	228.89	<0.0001
X1-Alginate	7528.63	1145.33	<0.0001	127.03	47.36	0.0002	14.73	17.33	0.0042
X2-PAG	1386.19	210.88	<0.0001	44.09	16.44	0.0048	0.06	0.07	0.7932
X1X2	1097.69	166.99	<0.0001	55.46	20.68	0.0026	500.90	589.11	<0.0001
X12	2477.60	376.92	<0.0001	52.73	19.66	0.003	470.70	553.59	<0.0001
X22	4.52	0.6877	0.4343	41.13	15.34	0.005	76.12	89.53	<0.0001
Lack of Fit	42.41	15.67	0.0112	18.03	31.94	0.003	1.62	5.98	0.05

**Table 4 pharmaceutics-14-00916-t004:** Composition, % yield, particle diameter, and encapsulation efficiency of TRHC-PAG-alginate microspheres.

Formulations	Components	D-P Ratio (g)	CaCl_2_ W/V (%)	Yield (%) Mean ± SD	Diameter (µm) Mean ± SD	EE (%) Mean ± SD
TA-1	TRHC-ALG	1:6	10	88.34 ± 0.96	657.67 ± 08.74	77.47 ± 0.69
TPB-2	TRHC-ALG-PAG	1:4:2	10	86.60 ± 1.14	633.42 ± 10.59	81.78 ± 0.42
TPB-3	TRHC-ALG-PAG	1:3:3	10	91.27 ± 0.85	607.34 ± 06.02	83.74 ± 0.79
TPC-4	TRHC-ALG-PAG	1:4:2	10	82.13 ± 1.54	591.54 ± 09.01	78.59 ± 0.67
TPC-5	TRHC-ALG-PAG	1:3:3	10	80.46 ± 1.08	579.23 ± 07.09	65.86 ± 0.26

TPB: Blended formulation; TPC: Coated formulation; DP: Drug Polymers; EE: Encapsulation efficiency; TRHC: Tramadol hydrochloride; ALG: Sodium alginate; PAG: Prunus armeniaca gum; *n* = 3.

**Table 5 pharmaceutics-14-00916-t005:** Kinetic evaluation of the in vitro drug release data of various TRHC-PAG-alginate microspheres.

Formulation	Correlation Coefficient (R^2^)				
	TA-1	TPB-2	TPB-3	TPC-4	TPC-5
Zero-order	0.8953	0.9451	0.9285	0.8569	0.8467
First-order	0.7762	0.8595	0.8476	0.7727	0.7689
Higuchi	0.5985	0.6659	0.6411	0.5608	0.5519
Hixson-Crowell	0.8139	0.8886	0.8746	0.7991	0.7934
Korsmeyer-Peppas	0.9803	0.9954	0.9966	0.9956	0.9917
Release exponent (*n*)	1.604	1.434	1.537	1.985	2.041

**Table 6 pharmaceutics-14-00916-t006:** Observations (Mortality, illness sign, food intake, water intake, and body weight) in control and TRHC-PAG-alginate microspheres treated group.

Parameters to Observe	Control Group	Treated Group
Mortality	Nil	Nil
Sign of illness	Nil	Nil
Food intake (g)		
Pre-treatment	2.8 ± 1.4	2.4 ± 1.2
Day 1	2.9 ± 1.6	2.7 ± 1.4
Day 3	3.2 ± 1.4	3.0 ± 1.6
Day 7	3.8 ± 1.0	3.6 ± 1.2
Day 10	4.0 ± 1.4	3.8 ± 1.4
Day 14	4.4 ± 1.2	4.2 ± 1.6
Water intake (mL)		
Pre-treatment	6.8 ± 1.4	6.4 ± 1.0
Day 1	7.4 ± 1.2	7.2 ± 1.4
Day 3	8.2 ± 1.6	7.8 ± 1.6
Day 7	10.2 ± 0.9	10.4 ± 1.2
Day 10	10.6 ± 2.0	10.9 ± 1.8
Day 14	12.2 ± 1.0	12.6 ± 0.9
Bodyweight (g)		
Pre-treatment	22.5 ± 1.4	22.8 ± 1.2
Day 1	23.8 ± 1.2	24.2 ± 1.4
Day 3	24.5 ± 1.6	24.9 ± 1.2
Day 7	27.5 ± 1.2	27.9 ± 1.6
Day 10	29.8 ± 1.4	30.2 ± 1.2
Day 14	31.5 ± 1.6	31.8 ± 1.4

**Table 7 pharmaceutics-14-00916-t007:** Biochemical analysis of control and TRHC-PAG-alginate microspheres treated group.

Plasma Analysis	Control Group	Treated Group
ALT (IU/L)	62 ± 1.9	58 ± 2.3
AST (IU/L)	136 ± 5.8	124 ± 6.3
Cholesterol (mg/dL)	127 ± 4.3	122 ± 4.2
Triglyceride (mg/dL)	126 ± 3.8	116 ± 3.4
Creatinine (mg/dL)	0.33 ± 1.1	0.36 ± 0.9
Urea (mg/dL)	60 ± 2.2	58 ± 2.4
Uric acid (mg/dL)	5.4 ± 1.6	4.8 ± 1.8

**Table 8 pharmaceutics-14-00916-t008:** Absolute weight of different vital organs (g) in control and TRHC-PAG-alginate microspheres treated group.

Group	Liver	Stomach	Heart	Kidney	Spleen
Control	5.86 ± 0.16	1.56 ± 0.44	0.58 ± 0.03	0.86 ± 0.03	0.62 ± 0.01
Treated	5.66 ± 0.18	1.62 ± 0.64	0.52 ± 0.05	0.79 ± 0.04	0.58 ± 0.02

**Table 9 pharmaceutics-14-00916-t009:** Analgesic activity evaluation of TRHC-PAG-alginate microspheres.

Groups	Number of Writhes (Mean ± SEM)	Reduction in Writhes Count (%)
Control (2.5 h prior administration of non-drug loaded PAG-alginate microspheres) + (1% acetic acid)	62.4 ± 5.8	0%
Treated (2.5 h prior administration of TRHC drug-loaded PAG-alginate microspheres) + (1% acetic acid)	30.8 ** ± 4.2 ** *p* < 0.01 (*n* = 6)	50%

## Data Availability

Not applicable.

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
