# Peer review of "Prunus armeniaca Gum-Alginate Polymeric Microspheres to Enhance the Bioavailability of Tramadol Hydrochloride: Formulation and Evaluation"

_pharmaceutics, 2022, doi:10.3390/pharmaceutics14050916_

Round 1
Reviewer 1 Report
The authors require to make improvements that should be addressed before its publication. The next major revisions to address are:
- In the abstract, need to mention the delivery is inside stomach, thus the very acidic pH selected.
- Introduction: More linkers to connect paragraphs will help the reader.
- Materials and methods: please combine techniques in same paragraph, such as thermal analysis (TGA and DSC).
- Table 1. Better to be moved to experimental section. Also, here, too many decimals are written (e.g., 4.41421 g of ALG seems too precise)
- Table 2: what is the drug-polymer ration units? (w/w%?) Please, combine table 2 and 5 to be more clear for the reader.
- Swelling studies: the use of tea bags does not sounds very conventional, can you explain this?
- Section 2.16. Ethics about the mice use should be also added before References, to be more easy to read.
- Results and discussion: Make a drawing/scheme of the microspheres preparation (like you did in the graphical abstract of your previous paper https://doi.org/10.1016/j.ijbiomac.2018.01.058)
- All the graphics and figures in general are distorted. Please, make them more professional (the numbers are too little, so no readable, and the figures seems all stretched. The presentation of the results should be improve to be published.
- Figure 2. TGA first % cannot be 82.68%, maybe you meant 17.32% like you mentioned in the text? please modify it accordingly.
- Equations 1-3 are not clear explained. Please explain it better.
- Swelling studies were done in buffer solution, which type? PBS? Also, pH 7.4 is not alkaline, but neutral, please change it in the text.
- Figure 6. Which pHs did you plot here? Please, add both release studies at both pHs, otherwise one cannot consider the release is better or higher at acidic pH from gastric acids, for bioapplications.
- FTIR is plotted wrongly, please do base line correction or any further needed modifications.
- Figure 8a. Why the drug signal is not visible for the blended microspheres? explain that.
- section 3.12. and fig 9a: It is not DSE but DSC. Please add in figure a, b or c, corresponding to each line and material. Legend is not enough.
- Section 3.13. The microspheres seems not too homogeneous. Please, add more SEM images showing the spheres.
- Which units did you use in Table 7? g, Kg, L....?
Author Response
We appreciate the suggestion and comments from the reviewer and have tried our best to incorporate the changes in the manuscript with tracking changes turned on. The language and format of manuscript have been modified. The mistakes and errors you suggested have been corrected. Please see the attached document for point-by-point response.

Reviewer 2 Report
The manuscript presented for review shows the results of research on obtaining polysaccharide hydrogel microspheres from a blend made of alginates and Prunus Armeniaca Gum, and the use as a matrix system during tramadol hydrochloride release.
The work is not particularly revealing as similar drug release systems have been described previously. However, it is by no means a repetition of the research, I have not come across information on the formation of blends with the composition proposed by the authors. In contrast, Prunus Armeniaca Gum was previously used to release curcumin (https://doi.org/10.1002/fsn3.2562).
The work is quite extensive, it describes in detail the individual issues related to the optimization of the composition, properties and conditions of microspheres formation. Describes the sustained release effect of a water-soluble drug. Additionally, the authors introduced the results of toxicological tests, but I am not convinced of the necessity of performing tissue morphology tests.
Generally, I believe that the work can be accepted for publication. However, the authors should make at least a few additions.
- Introduction - there are lack of literature references and no short description showing previous results of research on the release of drugs from microspheres formed from very similar polysaccharides or blends of polysaccharides forming hydrogels (e.g. Materials Science & Engineering C 104 (2019) 109958), as well as the use of Prunus Armeniaca Gum for this purpose.
- The authors posted the results of an acute toxicology study conducted on an animal model, whether the authors had obtained appropriate approval for such studies from a bioethics committee?
- Figure 3. - The purpose of these microscope photos is quite incomprehensible, what are they supposed to indicate? A broader description of the images shown is needed.
Formulation optimization - This chapter should be expanded with more comprehensive explanations. Indicate please the calculation methodology used, the source on which the created equations, theory and methodology were based. The name of the software used is not sufficient. Please explain the decision to choose a selected mathematical (polynomial) model, the course of estimating the degree of dependence of parameters, etc. How did the assumed model make it possible to shorten the optimization tests in practice?
Figure 6 - The drug release times seem a little too short from the application point, how can they be extended? Please comment at work.
Figure 10 - SEM images showing changes in the morphology of the microspheres with the release time of the drug, e.g. after 1-3 h and 12 h, should be inserted
There are no basic studies or data on the degradation of the presented hydrogel. This may be important when attempts are made to administer this material by any route other than the oral route. Please comment.
By what route can this release system be administered, or by oral administration only? What positive and evident changes were observed in the release from the matrix formed from the ALG + PAG blend compared to the matrix composed only of ALG. I hardly see it, please explain it in more detail in the Conclusions part.
Author Response
We appreciate the suggestion and comments from the reviewer and have tried our best to incorporate the changes in the manuscript with tracking changes turned on. The language and format of manuscript have been modified. The mistakes and errors you suggested have been corrected. Check the attached document please for point-by-point comment response

Round 2
Reviewer 1 Report
The authors have improved the manuscript, however still some improvements are required before its publication. The next revisions to address are:
Line 84. PAG nonparties --> nanoparticles
Swelling studies: again, the use of tea bags does not sound very conventional, can you explain this?
Figure 2a: FTIR spectra should have marked the assigned signals of the attributed chemical groups explained in the text. Also, this plot seems not professional and dodgy. Please make it according to scientific standards.
Also Figure 7, FTIR, it is again not appropriately plotted. One cannot read anything there, as it is plotted now.
Figure 5. 2D contour graphs: the numbers and legends are not readable. This cannot be published like that.
Author Response
We appreciate the suggestion and comments from the reviewer and have tried our best to incorporate the changes in the manuscript. The point to point response to reviewer is attached herewith.

Reviewer 2 Report
Most of my postulates were taken into account by the authors. However, there are still two matters not fully clarified.
- It is not entirely true that it was not possible to observe the morphology of the microspheres with the SEM after 2h and 12h. After this time, the samples can be stored in a frozen form; in such a state practically nothing happens to them. Before analysis, they can be quickly thawed and dried.
- There are still no results showing the decrease in microsphere mass during the drug release observations. There is only a comment in the text; it must be supported, for example, with an appropriate drawing showing the dependence of the change in the mass of microspheres on time, similar to the drawing showing the changes in the swelling index.
Author Response

(The authors gave the same response as above.)
